# A Maturity Model for Resilient Safety Culture Development in Construction Companies

**Minh Tri Trinh** [1] and **Yingbin Feng** [2,*]

1   Department of Civil Engineering, Mien Trung University of Civil Engineering, 24 Nguyen Du,
    Tuy Hoa 56000, Vietnam; trimitri0605@gmail.com
2   School of Engineering, Design and Built Environment, Western Sydney University,
    Locked Bag 1797, Penrith, NSW 2751, Australia
*   Correspondence: y.feng@westernsydney.edu.au

**Abstract:** A resilient safety culture is characterized by the capability of addressing the changing and unforeseen safety risks associated with the increasingly complex nature of sociotechnical systems, and creating an ultrasafe organization. An assessment of the maturation of resilient safety culture helps organizations to evaluate their capabilities of managing safety risks and achieving a consistently high safety performance. This study aims to present a maturity model developed to measure and improve resilient safety culture in the construction environment. The research was conducted in two stages. The first stage consisted of a review of the literature on the concepts of a resilient safety culture and the capability maturity model for the development of a maturity model. In the second stage, the developed model was evaluated using the Delphi technique. The model defines five maturity levels that can be used to measure resilient safety culture of a construction organization. It presents a set of descriptions of 19 aspects of resilient safety culture at each maturity level. The assessment procedure and the way of using the model are further discussed. Theoretically, this study provides insights into the maturity characteristics of a resilient safety culture. In practical terms, it offers guidance for benchmarking and encouraging the enhancement of organizations' capabilities to manage safety risks.

**Keywords:** construction; maturity model; resilience; safety; safety culture

## 1. Introduction

In recent years, despite substantial effort by many parties, the construction industry has been acknowledged as having inherent safety risks with high levels of change and uncertainty due to the increasing complexity of construction projects in terms of its technical, organizational and environmental factors [1]. For examples, a study by Albert et al. [2] conducted on diverse projects in the United States revealed that more than 50% of construction hazards remain unidentified. In the United Kingdom, Carter and Smith [3] revealed that up to 33% of hazards remain unrecognized in work method statements. Thus, the changing and unforeseen nature of safety risks poses challenges for construction organizations to ensure a state of workplace safety [4].

Over the past four decades of the evolution of occupational health and safety management, safety culture has been recognized as a crucial approach for improving the safety performance of construction organizations [5–8]. Safety culture reflects the ability of individuals or organizations to deal with risks and hazards so as to avoid damage or losses but still achieve their goals [9]. To promote a high level of safety culture, construction organizations have adopted diverse and holistic safety strategies, which emphasize (1) creating a safety knowledge database; (2) assuming that all accidents are preventable and unacceptable; (3) improving safety management systems to identify, assess and control hazards; (4) extending safety management matters to the entire supply chain and involving

all stakeholders; (5) promoting a strong commitment to safety among management; (6) establishing explicit accountability and authority for safety and rewarding safe behavior; and (7) shaping beliefs, attitudes and commitment of employees to achieve safe behavior [10]. These strategies are generally developed based on previous experiences and incident reports and the assessment of historical data about safety risks, thereby taking precautions against the accidents that have previously happened [11].

Although beneficial, traditional safety management and safety culture approaches have not completely addressed all types of safety risks encountered on construction sites. In fact, traditional approaches are institutionalized through plans, processes, procedures and policies for safety management, which are not readily and easily adaptable to the natural and inevitable changes in work being conducted, and the emerging and unforeseen safety risks being encountered [12]. They tend to become obsolete or deteriorate over time as a consequence of changes and uncertainties and thus leave organizations vulnerable to potentially disastrous failure modes and unforeseen kinds of safety risks [13].

While researchers and practitioners have grappled with these challenges, it is recommended that construction organizations should not only look to the past and set up safety measures to prevent known risks from appearing again but also establish the capability to address potential new forms of safety risks. In light of the above, building upon resilience engineering principles, a resilient safety culture has been recognized as a promising concept to address the changing and somewhat unpredictable forms of safety risks associated with the increasingly complex nature of sociotechnical systems and achieve an ultrasafe organization [4,14,15]. Trinh et al. [4] defined resilient safety culture as *an organization's psychological, behavioral, and contextual capabilities to "anticipate, monitor, respond and learn" to manage safety risks and create an ultra-safe organization* (p. 06018003-2). Previous studies indicated that the development of a resilient safety culture can enhance the organization's capabilities of addressing project hazards, human errors of workers and unexpected events, thereby allowing the organization to achieve consistently high safety performance in the construction industry [4,16]. Thus, it is necessary for organizations to obtain a clear understanding of the mechanisms by which a resilient safety culture can be created and assess their current maturity of resilient safety culture.

The maturity of a resilient safety culture reflects the sophistication of the way that safety management practices are implemented in order to address safety risks in the organization [4]. Therefore, to achieve a "desirable", resilient safety culture in an organization, a starting point can be the identification and assessment of its current safety management practices. Based on such assessment results, recommendations for improvement measures can be derived and prioritized to reach higher maturity levels of a resilient safety culture. Although a number of studies have been conducted to investigate resilient safety culture in order to conceptualize the concept of a resilient safety culture in aviation organizations [14], identify the indicators of a resilient safety culture in petrochemical plans [15] and examine the impact of a resilient safety culture in construction safety management [17], few have focused on the processes that an organization should have to achieve a mature or advanced status with regard to a resilient safety culture. Against this background, this study aims to assess the maturity of resilient safety cultures in the construction environment. Based upon the capability maturity theory and the theory of resilient safety culture, the specific objectives are (1) to determine and define the maturity levels of a resilient safety culture, (2) to identify the key processes for enhancing resilient safety culture and (3) to develop a model for construction organizations to assess their resilient safety culture maturity.

The next section presents the methods for the development of the model. The findings pertaining to the three objectives are then discussed to clarify the contribution to knowledge and practical implications. The article ends with a discussion of limitations and recommendations for future research.

## 2. Methods

Maturity models involve defining maturity stages or levels that assess the completeness of the analyzed organizations or processes through various sets of multi-dimensional criteria [18]. Based on many existing maturity models, it has been recognized that the maturity model of resilient safety culture should comprise various measurable components, which include: the criteria and subcriteria, maturity levels and rubrics. Therefore, the present research was conducted in two stages, as shown in Figure 1.

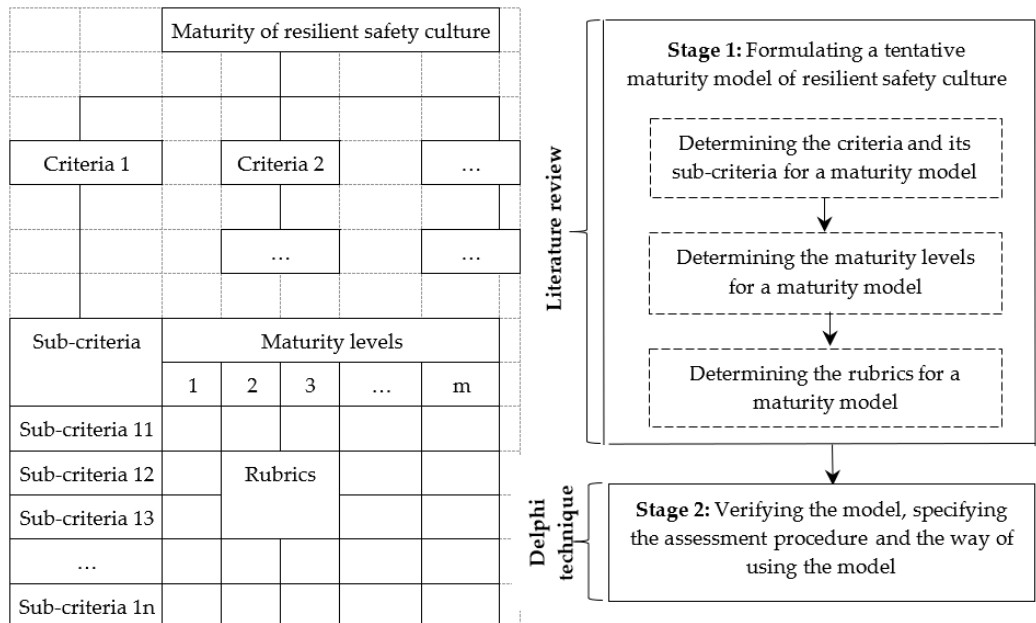

**Figure 1.** Structure of the maturity model of resilient safety culture and research process.

In the first stage, the literature review was conducted to (1) determine the criteria and their subcriteria, (2) select the number of maturity levels and (3) design the rubrics for the model of the maturity of resilient safety culture. A recent publication by Trinh et al. [4] is used as the starting point for the literature selection process. Trinh et al. [4] recognized that a resilient safety culture can be created in a construction organization by developing strategies and action plans following the principles of hazard prevention practice, error management practice and mindful organizing practice. Accordingly, hazard prevention, error management and mindful organizing are identified as the three main criteria of the maturity model of resilient safety culture in this study. To determine the subcriteria of each main criterion, the Scopus search engine was then used to identify scholarly work pertaining to such three criteria in the area of construction safety management. With regard to hazard prevention, the terms "hazard prevention", "construction" and "safety" were input in the title/abstract/keyword field of the Scopus search engine. In terms of error management practice, the terms "error management", "construction" and "safety" were used. In terms of mindful organizing practice, the terms "mindful organizing", "construction" and "safety" were used. Capability maturity theory and existing empirically verified maturity models were reviewed to determine the number of maturity levels and the form of the model. Finally, a detailed content review of the safety literature pertaining to the definitions and characteristics of each subcriterion was conducted to design the rubrics of the model.

To verify the developed model, the second stage of this study involves a Delphi method. The Delphi method is typically designed to collect the most reliable views from a group of experts through several rounds of intensive questionnaires interspersed with feedback in the form of controlled opinions [19]. In this study, the Delphi method is applied for the following reasons: (1) experts remain anonymous to one another; (2) it reduces the potential for influence or bias throughout the rounds; (3) it suits groups that

are geographically distant; (4) information and opinions are gained from a wide range of experts; and (5) the process ensures that experts are involved from the beginning, which can assist future policies or programs that may be developed from the results [20]. According to Ludwig [21], in order to obtain good research results, the majority of Delphi studies involve a panel of 15–20 respondents. In this study, two rounds of Delphi questionnaires with 15 experts were conducted. All participants were (1) actively working within the Australian construction industry with a minimum of 10 years' experience; (2) in a senior management or safety-management role; and (3) directly involved in safety management on construction projects. In the first round of the Delphi survey, respondents were asked to (1) provide their opinion on the selected criterion and its subcriteria and maturity levels and (2) answer open-ended questionnaires pertaining to the designed rubrics, which were specified in the first stage of this research. Accordingly, the survey was initiated by verifying the experts' designation, area of practice, qualifications, years of experience and any further details relevant to their industry experience. Then, the experts were required to rate, on a scale of 1 (low) to 5 (high), the comprehensiveness of the model, the objectivity of the model, the practicality of the model, the replicability of the model, the reliability of the model and the overall suitability of the model. Following the rating, respondents were requested to provide further comments on the model. The model was revised based on experts' feedback. In the second round of the Delphi survey, experts were asked to reassess the revised maturity model in light of the consolidated results obtained in the first round of the survey.

## 3. Results

### 3.1. Experts' Views Regarding the Maturity Model of Resilient Safety Culture

Regarding the comprehensiveness of the model, all experts shared the view that the model is clear, concise and easily understandable (4.6 out of 5). Experts commented that *"the model is comprehensive and covers all key aspects required to measure measuring safety resilience within a construction organization"*, and *"the model is thorough, detailed and adequate to describe levels of maturity, and thus can be adopted to provide an accurate assessment of an organization's maturity status"*. In relation to objectivity, there was a consensus that the model provided neutral and impartial statements (5 out of 5). Some points of feedback were that *"this model will enable results which are objective and without influence from beliefs, values or experiences"* and that *"the model was clear, succinct and without ambiguity, which would allow for use by smaller builders through to tier one contractors"*. A common view amongst the experts in regard to practicality of the model was that both upper management and employees are able to use it in order to follow and provide an evaluation (4.6 out of 5). An example is *"the practicality of the model in enabling it to be used by not only upper managers to assess where the business scores in relation to safety, it is also practical enough for employees to follow, which in turn enables them to assess and provide feedback to the business in relation to its resilience of safety culture and where they feel it is heading"*. With respect to the replicability of the maturity model, a mutual agreeance was made by the experts (4.6 out of 5). Some of the experts' ideas included that of *"the ability to use the model as an effective monitoring tool, the ability for its use as a constant monitoring tool of organizations growth"* and the fact that *"the model could be used as an audit tool, and went further to specifically comment on its ability to perform well as a live document to which amendments and additions can be made at any time as an organization matures"*. With regard to the reliability of the model, the experts considered the criteria and subcriteria to have good detail, ensuring that reliable results are obtained (4.6 out of 5). Accordingly, all experts expressed their belief in the reliability of the model based on its ability to demonstrate to organizations what areas require improvement and how these improvements can be obtained. Taken together, the overall evaluation of the conceptual maturity model is positive in the evaluation of the model's suitability for assessment, profiling and benchmarking capabilities for construction work safety (4.8 out of 5). Some experts added that *"the model will prove an excellent tool for benchmarking, with the ability to actively oversee and monitor safety within organizations"*, *"the model will provide

*assistance and understanding of what is necessary and required to improve resilient safety culture within the organization"*, *"model can be incorporated into organizations in varying ways, as a wholistic assessment on an organization or assessment of individual projects"*, and *"the model can be adopted into safety management within other industries"*. These results indicate that the model would be an effective tool for benchmarking and encouraging the enhancement of organizations' capabilities to manage safety risks. The validated model is presented in the next section.

### 3.2. Maturity Model of Resilient Safety Culture

#### 3.2.1. Criteria and Subcriteria

The criteria and subcriteria that identify the stages of maturity of resilient safety cultures in organizations were chosen based on a review of the literature pertaining to the concept of a resilient safety culture. According to Trinh et al. [16], the concept of a resilient safety culture had its theoretical foundation in safety culture theory and resilience engineering theory. Since the early 1980s, "culture" has been recognized as an essential concept to provide insights into the complex features of an organization. Organizations own their history and shared leadership and learning, which shape the attitudes and behaviors of their members [22]. Organizational culture reflects shared behaviors, beliefs, attitudes and values [23]. It also facilitates shared interpretations of situations and renders coordinated actions and interactions possible and meaningful [24].

Schein [22] theorized that an organizational culture progresses in three stages of evolution: Founding and Early Growth (e.g., the assumptions are created by founders of the organization), Midlife (e.g., the assumptions are socialized) and Maturity/Decline (e.g., the shared assumptions are continually held strongly within organizations) [22]. In line with Schein's [22] study, Westrum [25] developed a typology model, which characterizes three stages of advancement of organizational culture, namely pathological, bureaucratic and generative. At the pathological level, information is hidden, messengers are "shot", responsibilities are shirked, bridging is discouraged, failure is covered up and new ideas are actively crushed. At the bureaucratic level, information may be ignored, messengers are tolerated, responsibility is compartmentalized, bridging is allowed but neglected, organization is just and merciful and new ideas create problems. At the generative level, information is actively sought, messengers are trained, responsibilities are shared, bridging is rewarded, failure causes inquiry, and new ideas are welcomed [25]. The work of Westrum [25] was further extended by Hudson's [26] study, in which the "bureaucratic" stage is replaced by the "calculative" stage, and two extra stages (i.e., "reactive" and "proactive") are introduced. This innovation is favorable for providing more accurate classification and increasing the accessibility of the framework to industrial practitioners [27].

Organizational culture can be used as a framework to understand how values, attitudes and beliefs about safety work are expressed and how they might influence directions that organizations take with respect to safety culture [28]. Safety culture is therefore often acknowledged as a subset of organizational culture, where the beliefs and values refer specifically to matters of health and safety [29]. A review of the safety culture literature by Wiegmann et al. [30] identified a set of critical features regardless of the particular industry from the various definitions of safety culture. These critical features include the following: (1) shared values; (2) concern with formal safety issues and the management and supervisory systems; (3) involvement of all members; (4) impacts on employees' work behavior; (5) the safety culture being reflected in the organization's policies, procedures and systems; (6) the safety culture being reflected in an organization's willingness to learn from errors, incidents and accidents; and (7) endurance, stability and resistance to change [30].

Resilience engineering has been proposed as a potential solution to address the limitation of traditional safety management and safety culture approaches in responding to the changing and somewhat unpredictable forms of safety risks related to the increasingly complex nature of sociotechnical systems [31]. A review by Bergström et al. [13] summarized two interconnected lines of reasoning for resilience engineering: (1) resilience engineering

is an increasingly adopted concept to cope with the growing complexity of socio-technical systems, and (2) resilience engineering is considered as an approach to address inherent risks and hazards that emerge from this increasing complexity. The growing complexity in these systems leads to potentially disastrous modes of failure and new shapes of safety risks, thereby forming a need for resilience engineering [13,32,33]. The proponent of resilience engineering recognizes that an accident can be prevented by developing an organization's capability to create foresight and recognize and anticipate the changing forms of risks before adverse consequences occur [34]. A resilient organization, therefore, manages safety risks proactively and creates safety based on four principles (or capabilities): anticipating, monitoring, responding and learning [35].

Based on safety culture theory and resilience engineering theory, studies [14,15,36,37] have advocated the concept of resilient safety culture for safety management. A number of researchers have characterized various theoretical approaches and methods to assess the resilient safety culture in different sectors [15,36,37]. Shirali et al. [15] viewed culture as "an engineered organization" and thereby support the inclusion of several aspects of an organization to describe the components of a resilient safety culture. As a result, thirteen indicators representing the resilient safety culture were identified: competency, involvement of staff, accident investigation, safety management system, awareness, flexibility, management commitment, reporting culture, preparedness, risk assessment, learning culture, management of change and just culture. The results of Shirali et al.'s [15] study enable the managers and policymakers to identify current weaknesses relating to resilient safety culture in their organizations. The study by Trinh et al. [4] also indicates that a resilient safety culture could be created in a construction organization by systematically responding to the potential threats against which resilience protects. They include project hazards (regular threats), human errors (irregular threats) and unexpected failures (unexampled events) in the construction environment. Accordingly, it is suggested that a resilient safety culture can be developed by implementing hazard prevention, error management and mindful organizing practices [4]. A review of safety literature by Trinh [38] further identified 19 safety interventions with regard to hazard-prevention practice (10 safety interventions), error management practice (4 safety interventions) and mindful organizing practice (5 safety interventions) to enhance a resilient safety culture. These three key safety practices and their corresponding safety interventions also underwent assessment to ensure their internal consistency, reliability, and convergent and discriminant validity in a recent study by Feng and Trinh [39].

In this study, hazard prevention, error management and mindful organizing practices, along with the corresponding safety interventions, were used as three key criteria and subcriteria for the maturity model of resilient safety culture and were further confirmed through the Delphi technique. The analyses of the Delphi study indicate that (1) all experts agreed that the three main criteria are sufficient to characterize resilient safety culture in the construction environment and (2) all experts believed that the subcriteria pertaining to its main criteria were acceptable.

### 3.2.2. Maturity Levels

In this study, the maturity levels for a model of maturity of a resilient safety culture were determined based on capability maturity theory and a review of existing empirically verified maturity models to assess safety issues. The concept of a capability maturity model was developed in the software industry by Philip Crosby, as referenced in Wendler [18]. Crosby theorized that software organizations undergo five successive stages of quality maturity in order to achieve the maximum level of quality, namely uncertainty, awakening, enlightenment, wisdom and certainty [18]. While the management in the uncertainty stage has no comprehension of quality as a management tool, there are transformations in management to achieve quality in the intermediate stages with respect to (1) how quality appears within an organization, (2) how organizational problems are handled, (3) the cost of quality as a percentage of sales, (4) quality improvement actions taken by management

and (5) how management summarizes the organization's quality problems. The certainty stage recognizes quality management as a vital part of the company [18]. Accordingly, a capability maturity model is developed to provide guidance for organizations in choosing process improvement strategies through the determination of current process maturity and the identification of the most critical issues for the process improvement [40]. Paulk et al. [41] summarized the five capability maturity levels as follows:

1 Initial (level 1): The process is described as ad hoc and is occasionally chaotic. Few processes are defined, and success depends on individual effort.

2 Repeatable (level 2): The project management process is developed to track cost, schedule and functionality. The process disciplines are used to assist a repeatable success on similar projects.

3 Defined (level 3): Both management and engineering activities are documented, standardized and integrated into a standard process. The standard organization processes are then applied to all projects.

4 Managed (level 4): The process and product quality are collected and measured in order to be quantitatively understood and controlled.

5 Optimizing (level 5): The process is continuously improved through quantitative feedback and innovative ideas, skills and technologies.

Capability maturity theory has been adopted to develop maturity models to assess safety issues across a wide range of industries (e.g., construction, oil and gas and healthcare). A review by Goncalves Filho and Waterson [40] indicated that the majority of maturity models in occupational health and safety research were formulated based on the combination of capability maturity theory and Westrum's [25] "Typology of Organizations". Westrum [25] theorized that one method to distinguish between organizational cultures was according to the way that safety-related issues were handled in the organization and that the introduction of a revised safety management or top leadership might present increasing levels of advancement of organizational culture. Consequently, it has been observed that five-level models have been proposed and tested the most frequently. In level 1 (Pathological), safety is a problem caused by workers, and the main drivers are the business and a desire not to be caught by the regulator. In level 2 (Reactive), organizations start to take safety seriously but there is the only action after incidents. In level 3 (Calculative), safety is driven by management systems, with much collection of data. Safety is still primarily driven by management and imposed rather than looked for by the workforce. In level 4 (Proactive), with improved performance, the unexpected is a challenge. Workforce involvement starts to move the initiative away from a purely top-down approach. In level 5 (Generative), there is active participation at all levels of the organization. Safety is perceived to be an inherent part of the business. Organizations are characterized by chronic unease as a counter to complacency [42]. Accordingly, the five-level maturity model of resilient safety culture in this study was theoretically supported by the literature review and further confirmed through the Delphi study.

### 3.2.3. Rubrics

Rubrics mainly contain evaluative criteria, quality definitions for those criteria at particular levels of achievement and a scoring strategy described in a table format and used for assessment [43]. Rubrics are the core of the maturity model of resilient safety culture because they present a set of instructions, which can be used to measure the detailed subcriteria in terms of their different maturity levels. In this study, the rubrics were developed based on the review of safety literature pertaining to the definitions of each subcriterion and previous maturity models on health and safety [27,42,44–48]. The rubrics were then refined by conducting two rounds of Delphi questionnaires. The analyses of the Delphi study indicate that (1) all experts approved of descriptions of the resilient safety culture at five maturity levels, and (2) most experts believed that, as an organization may not assert that it has a specific maturity level of resilient safety culture without having passed through appropriate criteria of maturity of safety culture, it is acceptable that

all subcriteria exert an equal effect in improving resilient safety culture. The results are presented in Tables 1–3.

**Table 1.** Rubric of hazard prevention.

| Subcriterion | Description | Level 1 | Level 2 | Level 3 | Level 4 | Level 5 |
|---|---|---|---|---|---|---|
| Safety Policy ($H_1$) [28,49] | The written safety policy provides the specific safety requirements for a construction project, which include the extent to which safety is a priority, the degree to which employees are consulted on health and safety issues, and the practicality of identifying hazards and implementing safety plans, procedures and instructions. | The organization does not consider health and safety requirements as equally important as other objectives. | The organization recognizes the importance of health and safety requirements only after hazardous events occur. | The organization sets objectives for health and safety performance within the workplace. | Safety policy is only available to site management and supervisors. | Safety policy is available to all workers, reflecting management's concern for safety, principles of action and objectives to achieve. |
| Site Safety Organization ($H_2$) [50] | Outlines the structure of the organization and the individual safety responsibilities and presents an organizational chart. The aim of site safety organization is to ensure the compliance with WHS standards, codes and legislation. | The organization does not provide safety plans/procedures on site. | The safety plan/procedures are written focusing on hazardous situations that occur repeatedly. | The safety plan/procedures are written focusing only on observed hazards. | The safety plans/procedures are written for all areas in the workplace but not periodically reviewed. | The safety plans/procedures are for all areas in the workplace and constantly reviewed for better effectiveness. |
| Risk Assessment and Hazard Analysis ($H_3$) [51,52] | Can be initiated by examining the activities related to a construction process, recognizing potentially hazardous situations that can result in an injury and assessing the probability and severity of all hazards for a specific activity. The implementation of hazard analysis and risk assessment can offer contractors an identification of the risk level of construction activities, thereby allocating safety measures in a more efficient manner. | The organization does not produce an analysis of potential hazards and the risks of accidents. | The organization produces an analysis of potential hazards and their risks of accidents only after hazardous events occur. | The organization produces an analysis of potential hazards and their risks of accidents only for observed hazards. | The organization produces an analysis of potential hazards and their risks of accidents on an ongoing basis for all areas at the workplace. | The organization produces an analysis of potential hazards, potential changes in working conditions and their risks of accidents for all areas at the workplace. |
| Safety Inspection ($H_4$) [49,53] | Refers to the identification of hazardous conditions for the modification of such conditions as appropriate and/or at regular intervals. A safety inspection aims to identify uncontrolled hazardous exposures to the construction workers, violations of safety standards or regulations, or unsafe behaviors. | The organization does not conduct safety inspections of the workplace. | The organization conducts safety inspections only after hazardous events occur. | The organization conducts safety inspections only for observed hazards. | The organization conducts regular safety inspections for all areas in the workplace. | There is a formal system (technical and human resources) for ongoing monitoring of whether employees perform work safely and the status of the work environment. |

**Table 1.** *Cont.*

| Subcriterion | Description | Level 1 | Level 2 | Level 3 | Level 4 | Level 5 |
|---|---|---|---|---|---|---|
| Hazard Control Program ($H_5$) [50,54,55] | Aims to eliminate hazards using the process control before exposing workers to any adverse working conditions. | The organization does not provide financial, technical or human resources to achieve health and safety targets related to observed hazards. | The organization provides financial, technical and human resources related to observed hazards only after hazardous events occur. | The organization provides financial, technical and human resources only for observed hazards. | The organization provides financial, technical and human resources to achieve health and safety targets related to both observed hazards and potential hazards. | The appropriate preventive measures are immediately provided following any changes to the working conditions (i.e., new hazards identified, hazardous events occurred). |
| Personal Protection Program ($H_6$) [28] | The implementation of the personal protection program refers to the degree to which the organization is concerned with designing, issuing, using, and enforcing and monitoring PPE. | The organization does not provide any personal protective equipment (PPE) at work. | The organization provides PPE only after serious hazardous events occur. | The organization provides PPE only when required | The organization provides PPE complied with safety plans. | The organization provides and maintains PPE and inspects them for their proper use. |
| Safety Meetings ($H_7$) [56] | In these meetings, communication and information sharing are associated with the frequency and methods of emphasizing knowledge and the importance of safe work (e.g., informing potential hazards in the workplace, new or revised work instructions and safety rules, work tasks, and safety incidents experienced by other employees or organizations). | The organization does not organize any safety meetings at work. | The organization organizes safety meetings only after serious hazardous events occur. | The organization organizes formal safety meetings focusing only on observed hazards. | The organization organizes formal safety meetings on most of the safety-related issues. | The organization organizes formal safety meetings on all of the safety-related issues. |
| Safety Training ($H_8$) [57] | All workers should be provided with safety training about the hazards related to their work tasks. | The organization does not provide any safety training at work. | The organization provides a specific safety training program at work only after serious hazardous events occur. | The organization has a standard safety training program only for the employees who work in places where safety risks are identified. | The organization has a safety training program at work for all employees. | The organization has a continuous safety training process at work for all employees. |

**Table 1.** *Cont.*

| Subcriterion | Description | Level 1 | Level 2 | Level 3 | Level 4 | Level 5 |
|---|---|---|---|---|---|---|
| Safety Promotion ($H_9$) [49,55] | Includes promoting safety behavior and engaging employees in decision-making processes by implementing rewards/punishments, developing an advertising campaign (e.g., safety posters and stickers) or consulting them about their wellbeing. A well-designed safety promotion program is characterized by a high visibility level in the organization and offering recognition. The use of safety promotion can enhance reporting hazards, awareness and self-protection action among workers. | The organization does not have an incentive (reward or punishment) system in the WHS area. | The organization adopts an incentive system to stimulate safety at work only in specific situations, that is, after serious hazardous events occur. | The organization adopts an incentive system to improve safety performance only for those sectors where risks of hazards are identified. | The organization adopts an incentive system for all its sectors in order to improve safety performance of employees. | A provided incentive system is not necessary as employees are highly motivated to act safely. |
| Management Support ($H_{10}$) [54,58,59] | Safety support from the management is an observable activity on the part of the management support and must be demonstrated via their behaviors and words. | The management and supervisors do not give support in safety. | The management and supervisors give support only when hazardous situations occur. | The management and supervisors give support when health and safety issues are encountered. | The management and supervisors actively seek to find health and safety issues and provide sufficient support to employees at work. | The support offered by the management and supervisors is not necessary as everyone on site has a clear understanding of their roles and responsibilities in order to eliminate or reduce the risks of hazards |

**Table 2.** Rubric of error management.

| Subcriterion | Description | Level 1 | Level 2 | Level 3 | Level 4 | Level 5 |
|---|---|---|---|---|---|---|
| Learning from Errors ($E_1$) [60] | Aims to reduce the repeated errors or the adverse outcomes of errors in the future. Learning occurs when people are encouraged to learn from errors, when they think about errors meta-cognitively and when the negative emotional impact of errors is reduced. | Employees consider errors as not useful to improve safety performance on site. | Employees are concerned about specific errors only when accidents occur. | A minority of employees are concerned about how to avoid and/or correct errors. | The majority of employees readily accept feedback about how to avoid and/or correct errors. | All employees actively ask others for advice on how to avoid and/or correct errors. |

**Table 2.** *Cont*.

| Subcriterion | Description | Level 1 | Level 2 | Level 3 | Level 4 | Level 5 |
|---|---|---|---|---|---|---|
| Error Competence ($E_2$) [61] | Refers to knowledge or capability of individuals to deal immediately with errors. | When errors are made, employees ignore it them carry on with their work tasks. | When errors are made, employees are interested in correcting them only when accidents occur. | When errors are made, a minority of employees are engaged in correcting errors. | When errors are made, the majority of employees are engaged in correcting errors. | When errors are made, all employees on site are engaged in correcting it. |
| Thinking About Errors ($E_3$) [62] | Errors are used for exploration and experimentation in order to develop a better and more sophisticated understanding of a particular situation that caused an error to occur. | The organization does not analyze errors. | The organization analyzes errors only when those errors lead to accidents. | When errors are made, only management are concerned about them so they may be analyzed to identify the employee(s) at fault. | When errors are made, the majority of employees are interested in understanding how to avoid and/or correct them. | All errors are analyzed thoroughly by all employees in order to prevent their occurrences in the future. |
| Error Communication ($E_4$) [63,64] | Error communication refers to individuals' decisions to talk openly about errors to co-workers and supervisors or report through the official incident-reporting systems. Due to error communication, the knowledge from error learning allows workers to detect and deal with errors in hazard situations effectively. | The employees do not share or report any errors that occurred as they do not feel comfortable enough. | The employees share or report errors that occurred only when those errors lead to accidents. | The employees share or report errors that occurred only when they did not contribute to the occurrence of such events. | The employees share or report errors occurred, even if they contributed to the occurrence of such events. | All employees on site feel free to share errors with others and report to the organization so that the same mistakes do not occur again. |

**Table 3.** Rubric of mindful organizing.

| Subcriterion | Description | Level 1 | Level 2 | Level 3 | Level 4 | Level 5 |
|---|---|---|---|---|---|---|
| Preoccupation with Failure ($M_1$) [65,66] | Preoccupation with failure refers to directing attention and effort to a proactive and pre-emptive analysis of potential new sources of conditions that can produce the unexpected. This means that employees actively and continuously search for indicators of failure. | No one on site acknowledges that unexpected hazardous events (i.e., unobserved hazardous conditions and unintentional unsafe behaviors) can occur anytime and anywhere. | Employees are concerned about the unexpected only when accidents occur. | A minority of employees are mindful of safety risks on site even when they were recognized and controlled. | The majority of employees are mindful of safety risks on site even when they were recognized and controlled. | There is no sense of complacency about health and safety measures implemented on site. |

**Table 3.** *Cont.*

| Subcriterion | Description | Level 1 | Level 2 | Level 3 | Level 4 | Level 5 |
|---|---|---|---|---|---|---|
| Reluctance to Simplify Interpretations ($M_2$) [67] (p. 139) | Refers to "deliberately questioning assumptions and received wisdom to create a more complete and nuanced picture of current situations". Employees do not take the past as an infallible guide to the future but rather actively seek divergent viewpoints that question received wisdom, uncover blind spots and detect changing demands. | The organization does not appreciate when employees express their viewpoints on how to improve health and safety on site. | The organization is aware of the importance of discussion and exchange of views about safety risks only after accidents occur. | The organization readily accepts various viewpoints on how to improve health and safety on site. | The organization actively seeks various viewpoints on how to improve health and safety on site. | There is an open channel of communication within the organization to collect and collate various viewpoints on how to improve health and safety on site. |
| Sensitivity to Operations ($M_3$) [68] | Refers to creating and maintaining an up-to-date understanding of the distributed tasks and expertise so that these are appropriately utilized when the organization is faced with unexpected events. This requires (1) a strong contact between employees to make sure inconsistencies and problems are quickly recognized and treated and (2) a number of adjustments are made in order to prevent the compounding of failures. | The organization does not provide the employees with information on the hazards related to their work tasks before commencing work. | The organization only provides the employees with information on the hazards related to their work tasks when hazardous events occur. | The organization provides the employees with up-to-date information about safety risks to conduct work task safety before commencing the work tasks. | Employees on site actively seek comprehensive and complete information on the hazards related to their work tasks. | Employees interact often enough to build a clear picture of what is happening on site. |
| Commitment to Resilience ($M_4$) [67] | Refers to developing capabilities to cope with, contain and bounce back from mishaps that have already occurred before they worsen and cause more serious harm. | The organization does not prepare for unexpected events, and no one knows what to do in the cases of emergency situations (i.e., injury, damage to properties, incident). | The organization is aware of the importance of preparation for the unexpected only after unplanned hazardous situations occur. | The minority of employees react quickly to emergency situations (i.e., injury, damage to properties, incident). | The majority of employees react quickly to emergency situations (i.e., injury, damage to properties, incident). | All employees on site know what to do in the case of an accident at work, and they are prepared for unexpected events. |
| Deference to Expertise ($M_5$) [69] | Occurs when people with the best expertise in managing the problem at hand make decisions, regardless of their formal rank in the face of an unexpected event. | The organization does not obtain expert assistance when unfamiliar safety issues come up. | The organization seeks to obtain expert assistance only after an accident occurs. | When unfamiliar events occur, the management asks their employees for advice on how to resolve them. | When a health and safety issue out of the ordinary occurs, everyone on site knows who has the expertise to respond. | All employees on site have the expertise to respond to health and safety issues that may occur out of the ordinary. |

To examine the resilient safety culture of a particular construction organization (i.e., a project), each subcriterion referring to a case study can produce a score. In this study, as the weights or relative importance of criteria and their subcriteria have not been examined, the equal weights method [70], which requires minimal knowledge of the decision-maker's priorities and minimal input from the decision-maker, was employed. The final score for error management criteria is then determined as an average score of all its subcriteria (Equation (1)). Using the rubric of error management criteria as an example, for a specific subcriterion, if the rubric of a case study falls in the Pathological level, it scores 1; if it falls in the Reactive level, it scores 2; and so on. An example of Project A could be scored using Equation (1) and shown in Table 4. Accordingly, with regard to error management criteria, as the final score of Project A is 3.5, Project A is in the Proactive level.

$$\text{Final score of error management criteria}: \ S(E) = \frac{1}{4}\sum_{i=1}^{4}S(E_i) \tag{1}$$

where $S(E_i)$ is the score of the subcriteria $E_i$.

**Table 4.** Example of using the maturity model of resilient safety culture in Project A (error management criteria).

| Subcriterion | 1 | 2 | 3 | 4 | 5 | Final Score |
|---|---|---|---|---|---|---|
| $E_1$ | | ▓▓ | | | | 3.5 |
| $E_2$ | | | | ▓▓ | | |
| $E_3$ | | | ▓▓ | | | At Level |
| $E_4$ | | | | | ▓▓ | 4 |

The overall score of the maturity of resilient safety culture for a construction project can be determined as an average score of all three key criteria (Equation (2)). Accordingly, if the overall score of a resilient safety culture falls between 1 and 2, the maturity of resilient safety culture is at the Reactive level; if it falls between 2 and 3, the maturity of the resilient safety culture is at the Calculative level, and so forth. The maturity model of a resilient safety culture is illustrated in Figure 2.

$$\text{Overall score of resilient safety culture}: \ S = \frac{1}{3}[S(H) + S(E) + S(M)] \tag{2}$$

where $S(H)$ and $S(M)$ are the final scores of hazard prevention and mindful organizing, respectively.

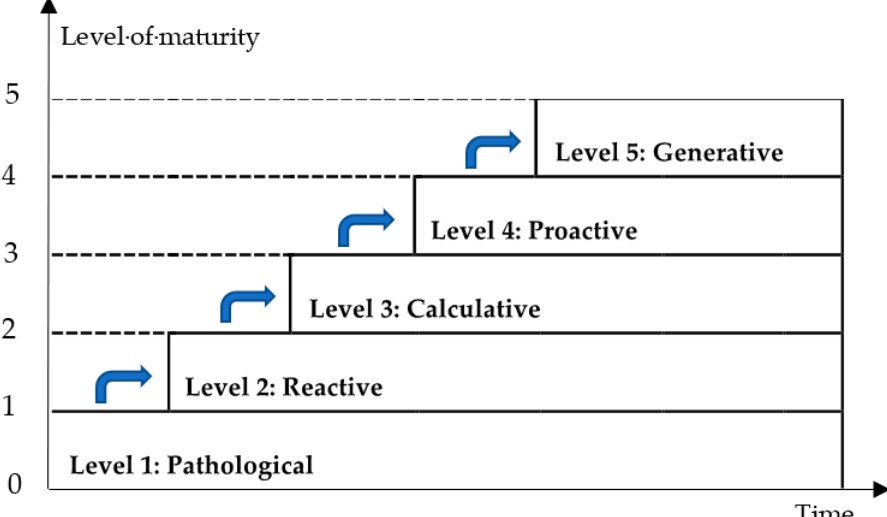

**Figure 2.** Resilient safety culture maturity levels.

Furthermore, once the same procedures are repeated in multiple projects for the same company, it is possible to sum them up and perceive the maturity level of the resilient safety culture at a company level. It has been noted that although there are various projects within the same company, such projects could have different levels of resilient safety culture maturity in terms of hazard prevention, error management and mindful organizing. This enables resources to be allocated more efficiently to achieve an advanced status with regard to resilient safety culture. Likewise, it is possible to obtain the maturity level of a group of similar companies or the whole industry. It therefore offers a means of benchmarking for resilient safety culture maturity and allows the required actions to be identified before the higher maturity can be achieved.

## 4. Discussion

This study expands the existing literature relating to resilient safety culture by proposing a quantitative maturity model of a resilient safety culture for construction organizations, thereby facilitating an understanding of developing the concept of a resilient safety culture in the construction industry. The model presented in this paper has the following key features:

1   It integrates three related concepts, namely hazard prevention, error management and mindful organizing practices, and uses these concepts as three main criteria to assess resilient safety culture. Three key criteria, therefore, allow the resilient safety culture to be observed and enhanced in different aspects when making an assessment.
2   It employs a five-level capability maturity model to measure resilient safety culture, thus allowing the level of resilient safety culture to be assessed through the proposed level in a range of "1 = Pathological" to "5 = Generative".

In practical terms, the maturity model of a resilient safety culture has two applications. The first application is to provide the employees with the perceptions with regard to its current safety management practices and the maturity level of resilient safety culture. It is therefore intended to be used in safety meetings or workshops to offer participants a clear view of the status quo, strengths and weaknesses of their organizations' capabilities to manage safety risks. Within the same company, managers can also use the model as a tool to compare construction projects with regard to their resilient safety culture maturity. The second application of the model is to provide guidance for the enhancement of resilient safety culture maturity. Based on the proposed model, it is clear that an organization can obtain different scores in terms of hazard prevention, error management and mindful organizing. Based on the assessment results of the proposed model, it is suggested that organizations can better recognize the specific areas and the safety practices required, and thereby allocate resources efficiently in order to achieve an advanced status with regard to resilient safety culture.

## 5. Conclusions

Resilient safety culture has been recognized as a promising concept to establish an ultrasafe organization. This paper reports the development of a maturity model for a resilient safety culture in the construction environment. Based on an extensive review of pertinent literature, the components for the maturity model were identified. The Delphi method was then employed to verify the model. As a result, the developed model consists of 5 maturity levels, 3 assessment criteria, and 19 detailed assessment subcriteria. Detailed descriptions of each assessment subcriterion at five different maturity levels are also presented. The proposed model is useful because it enables organizations to benchmark their current level of resilient safety culture maturity and identify the actions required before the higher maturity can be achieved.

This research has several limitations. First, the key criteria and their subcriteria for assessing and enhancing resilient safety culture have not been prioritized, and thus it is assumed that all subcriteria have the same weight. The organization using this maturity model could either apply the equations provided in this study or develop a scale of their

importance. Future research can focus on calculating the overall score of resilient safety cultures considering the weighted value of assessment subcriteria. The second limitation is that the proposed model was not tested using empirical data. It would be worthwhile to conduct a case study in order to demonstrate the application of the proposed model. For example, several construction projects under construction stage by a contractor can be selected for assessment and comparison.

**Author Contributions:** Conceptualization, M.T.T. and Y.F.; methodology, M.T.T. and Y.F.; validation, M.T.T. and Y.F.; formal analysis, M.T.T.; investigation, M.T.T.; data curation, M.T.T.; writing—original draft preparation, M.T.T.; writing—review and editing, Y.F.; visualization, M.T.T. and Y.F.; project administration, Y.F; funding acquisition, Y.F. All authors have read and agreed to the published version of the manuscript.

**Funding:** This research received no external funding.

**Institutional Review Board Statement:** The study was conducted in accordance with the National Statement on Ethical Conduct in Human Research 2007 (Updated 2018). Ethical approval for this study was granted by the Western Sydney University Human Research Ethics Committee (HREC Approval Number: H12639).

**Informed Consent Statement:** Informed consent was obtained from all subjects involved in the study.

**Data Availability Statement:** Data that support the findings of this study are available from the corresponding author upon reasonable request.

**Acknowledgments:** This research was supported by Western Sydney University through Research Development Funding, ACA/DAP Research Scheme Support.

**Conflicts of Interest:** The authors declare no conflict of interest.

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
