# Peer review of "A Maturity Model for Resilient Safety Culture Development in Construction Companies"

_buildings, doi:10.3390/buildings12060733_

Round 1

Reviewer 1 Report

1.P2 lines 87, what does the maturity of resilient safety culture stand for? And how do define the mature or advanced status? What are the differences in the resilience culture in different fields?

2.P3 lines 124, experts were required to rate, the rate is not shown to support the comprehensiveness, objective, practicality, replicability, and reliability of the model? Are the results of expert scoring evaluated for consistency and validity?  

3.P4 lines 172-, authors selected a literature review to develop the criteria and subcriteria. Please describe the literature selection process. Such as literature resources, principles of literature selection, and the number of selected literature.

4. Lines 180-209, whats the connection between the content and criteria. Lines 180-197 seem to be the statement of Rubrics. Suggest authors optimize the structure of the manuscript.

5.P5 lines 231, A number of researchers have characterized various theoretical approaches and methods to assess resilient safety culture in different sectors, What is different about the resilient culture in these fields?  What are the unique features of the resilient safety culture in construction?  Do the indicators identify later reflect these characteristics?  

6.P6 H2, please remove the underline of site safety organization

7.P6 some subcriterias has similar content, such as E1 and E3.

8. The revision of the model is not shown in the first stage.

9.P10 lines 336 font is not consistent.

10.P10 why think the wight of subcriteria is the same?

Reviewer 2 Report

Strengths of the paper:

This is a professionally laid out paper that is comprehensive, clear and solid in terms of the research methodology employed (literature review, Delphi Method) but while the scientific methods applied are simplistic (calculation of simple average) they are practical enough to allow ease of use of the proposed safety culture maturity model.

In my opinion, the merit of this paper is significant in that it contributes towards development of a safety culture maturity model to assess the safety culture in any organization that is developed based on results of extensive organizational research and is simple in application.

Weakness of the paper:

The most important weakness of this paper is already mentioned by the authors as a limitation, i.e. “that the proposed model was not tested using empirical data. It is worthwhile to conduct a case study in order to demonstrate the application of the proposed model.”

In my opinion, it is necessary to provide at least a full example of the application of the model and to carry out a sensitivity analysis of the results in order to prove the applicability and usefulness of the model.

A second significant weakness is that the full model is not presented. I believe you should include a table with all rubrics for all sub-criteria to help researchers interested in applying your model to real construction organizations.

Finally, the simplification that all criteria are equal needs to be justified scientifically or you should include in your model a step for determination of criteria weights by the evaluator.

I feel that your model, while a great idea, has not been calibrated to the needs of the construction industry. The inclusion of an example application is a minimum requirement and then you can restate that validation by application to a real construction organization should be carried out as further research.

Further Suggestions

1.     One aspect that hasn’t been considered is the different hazards and factors contributing to resulting accidents according to project type or construction phase. Can your proposed model be adjusted to consider the differences in safety culture according to construction field of the organization or project size?

See for example:

·         Antoniou F. and Merkouri M. (2021). Accident factors per construction type and stage: A synthesis of scientific research and professional experience, International Journal of Injury Control and Safety Promotion, DOI: 10.1080/17457300.2021.1930061

·         Berglund, L., Johansson, M., & Nygren. M., Samuelson, B., Stenberg, M., & Johansson J. (2019). Occupational accidents in Swedish construction trades. International Journal of Occupational Safety and Ergonomics, http://doi.org/10.1080/10803548.2019.1598123

·         Carrillo-Castrillo, J.A., Trillo-Cabello A.F., &  Rubio-Romero J.C. (2017). Construction accidents: identification of the main associations between causes, mechanisms and stages of the construction process. International Journal of Occupational Safety and Ergonomics, 23(2), 240-250.

  In the methods section

·         you have included detailed results from stage 2. These should be moved to a separate results section as the reader learns about the experts’ evaluation of the model before they learn about the model itself.

·         the justification of the use of the Delphi Method should be included here.

·         include an explanation of the scientific methods employed in your model which, is in essence an application of the simple additive weighting method (SAW) which is the most intuitive and easy way to deal with MCDM problems. It is actually a special case of multi attribute utility theory (MAUT).

See (Antoniou F., Konstantinidis D. and Aretoulis G. (2016) “Application of the multi attribute utility theory for the selection of project procurement system for Greek highway projects”, International Journal of  Management and Decision Making, 15(2), 83-112, DOI: 10.1504/IJMDM.2016.077761

The choice of criteria is a result of your literature review described in section 3.1. It would be better justified and evident to the reader if it were a result of the frequency of appearance in relevant research for the construction industry that could be depicted with a table with the references (columns) and the criteria (rows) and a check in the cell depicting that the particular criteria is included in . the particular . reference.

Finally, please see attached annotated version of the paper with additional few minor corrections.

Round 2

Reviewer 1 Report

Revision well done. It can be accepted as it is.

Reviewer 2 Report

All of my comments have been adequately addressed in the revised version.

I have no further comments.